# Evidence for Two Modes of Binding of the Negative Allosteric Modulator SB269,652 to the Dopamine D_2_ Receptor

**DOI:** 10.3390/biomedicines10010022

**Published:** 2021-12-23

**Authors:** Richard Ågren, Kristoffer Sahlholm

**Affiliations:** 1Department of Neuroscience, Karolinska Institutet, 17177 Stockholm, Sweden; 2Wallenberg Centre for Molecular Medicine, Department of Integrative Medical Biology, Umeå University, 90187 Umea, Sweden

**Keywords:** induced-fit binding, secondary binding pocket, allosteric modulation, G protein-coupled receptors

## Abstract

SB269,652 has been described as the first negative allosteric modulator (NAM) of the dopamine D_2_ receptor (D_2_R), however, the binding mode and allosteric mechanism of action of this ligand remain incompletely understood. SB269,652 comprises an orthosteric, primary pharmacophore and a secondary (or allosteric) pharmacophore joined by a hydrophilic cyclohexyl linker and is known to form corresponding interactions with the orthosteric binding site (OBS) and the secondary binding pocket (SBP) in the D_2_R. Here, we observed a surprisingly low potency of SB269,652 to negatively modulate the D_2_R-mediated activation of G protein-coupled inward-rectifier potassium channels (GIRK) and decided to perform a more detailed investigation of the interaction between dopamine and SB269,652. The results indicated that the SB269,652 inhibitory potency is increased 6.6-fold upon ligand pre-incubation, compared to the simultaneous co-application with dopamine. Mutagenesis experiments implicated both S193 in the OBS and E95 in the SBP in the effect of pre-application. The present findings extend previous knowledge about how SB269,652 competes with dopamine at the D_2_R and may be useful for the development of novel D_2_R ligands, such as antipsychotic drug candidates.

## 1. Introduction

The G protein-coupled dopamine (DA) receptors play important roles in neurological functions related to motor control, memory and cognition, reward-driven behavior, and endocrine regulation [1]. The dopamine D_2_-like family comprises of the dopamine D_2–4_ receptors (D_2_R, D_3_R, and D_4_R), which couple to Gα_i/o_ proteins, inhibiting adenylate cyclase and activating G protein-coupled inward-rectifying potassium (GIRK) channels and in addition, signal via arrestin-dependent pathways [2]. The current treatment of schizophrenia mainly relies on compounds which are antagonists and weak partial agonists at D_2/3_Rs. The older, first-generation antipsychotics are characterized by D_2/3_R antagonism and generally have a higher propensity for motor and endocrine side effects. These adverse reactions may be less common with newer second- and third-generation drugs due to action at additional sites including serotonin receptors and partial D_2/3_R agonism, respectively [3]. However, undesired collateral actions, as well as the suboptimal amelioration of cognitive deficits in schizophrenia, are still important limitations to existing therapy, thus warranting the development of novel therapeutic strategies [4]. One such strategy centers on the use of negative allosteric modulators (NAMs), which reduce DA potency and/or efficacy at D_2/3_Rs rather than compete with DA for binding to the receptor. This approach would provide the reduction of dopaminergic tone presumed relevant for antipsychotic efficacy, while preserving the physiological dynamics of DA signaling [5]. The first drug-like D_2_R/D_3_R NAM SB269,652 engages D_2/3_Rs in a bitopic manner, with its structure comprising a tetrahydroisoquinoline (THIQ) primary pharmacophore targeting the orthosteric binding site (OBS) between transmembrane helices (TMs) 3, 5, and 6, and an indole-2-carboxamide secondary pharmacophore engaging a secondary binding pocket (SBP) encompassing TM2 and extracellular loop 1 (ECL1) [6,7,8] (Figure 1A). The mechanism by which this ligand acts as an allosteric modulator despite interacting with the OBS remains controversial [5], but has been suggested to involve receptor–receptor modulation within D_2/3_R dimers, with SB269,652 binding only one protomer within such a dimer [7].

The affinities, efficacies, and (when applicable) allosteric properties of several D_2_/_3_R ligands encompassing both a primary and a secondary pharmacophore, including SB269,652, have been demonstrated to be dependent on the length of the linker joining these two pharmacophores [9,10]. Furthermore, mutations in TM2 of the D_2_R were shown to reduce negative cooperativity of SB269,652, indicating a role for the SBP in the allosteric actions of this ligand [7]. In agreement, compounds containing only the secondary pharmacophore behaved as allosteric ligands, whereas ligands consisting mainly of the primary pharmacophore were competitive antagonists [8]. However, interactions with the OBS have also been reported to modulate the allosteric properties of SB269,652, presumably by altering the positioning of the secondary pharmacophore in the SBP [11].

Taking advantage of the temporal resolution afforded by a GIRK channel activation assay, we recently demonstrated that the aripiprazole analogue SV-III-130 interacts with D_2_R by way of an induced-fit mechanism leading to the irreversible binding of this bivalent ligand. Furthermore, this behavior was observed to be crucially dependent on TM2 and ECL1 [9]. Here, using the same assay system, we investigated the influence of the OBS and SBP on the ability of SB269,652 to negatively modulate D_2_R activation by DA. We found evidence of an induced-fit (or possibly conformational selection) mechanism of binding of SB269,652, which is dependent on interactions with both OBS and SBP.

## 2. Methods

### 2.1. Molecular Biology

Wild-type (WT) human dopamine D_2L_ receptor (D_2_R) cDNA was in pXOOM (provided by Dr. Søren-Peter Olesen, University of Copenhagen, Copenhagen, Denmark). V91A, L94A, E95A, W100A, S193A, and S194A mutagenesis was performed by Genscript (Piscataway, NJ, USA). All mutations were verified by sequencing. cDNA encoding human GIRK1 (Kir3.1), GIRK4 (Kir3.4) (provided by Dr. Terence Hebert, University of Montreal, Montreal, QC, Canada) and regulators of G protein signaling 4 (RGS4) (from the Missouri cDNA Resource Center; www.cdna.org; accessed on 14 November 2021) were in pCDNA3 (Invitrogen, Waltham, MA, USA). Plasmids were linearized using the appropriate restriction enzymes (D_2_R, RGS4; XhoI and GIRK1/GIRK4; NotI), followed by in vitro transcription using the T7 mMessage mMachine kit (Ambion, Austin, TX, USA). cRNA concentration and purity were determined by spectrophotometry.

### 2.2. Oocyte Preparation

Oocytes from the African clawed toad, *Xenopus laevis*, were isolated surgically as described previously [12]. The surgical procedures were approved by the Swedish National Board for Laboratory Animals and the Stockholm Ethical Committee (approval number 686–2021). Following one day of incubation at 12 °C, oocytes were injected with 0.2 ng D_2_R cRNA, 40 ng of RGS4 cRNA, and 1 ng of each GIRK1 and GIRK4 cRNA using the Nanoject II (Drummond Scientific, Broomall, PA, USA) and a volume of 50 nl per oocyte.

### 2.3. Dopamine Receptor Ligands

DA and SB269,652 were purchased from Sigma-Aldrich (St. Louis, MO, USA). SB269,652 was dissolved in DMSO and further diluted in recording buffer, with a maximum final DMSO concentration of 1% *v*/*v* used in experiments. Ligand-mediated direct inhibition of GIRK currents (>10%) was not observed in control experiments where SB269,652 was applied to oocytes expressing GIRK1/4 in the absence of the D_2_R.

### 2.4. Electrophysiological Methods

Following RNA injection and 6 days of incubation at 12 °C, electrophysiological experiments were performed on the oocytes using the parallel eight-channel, semi-automated, two-electrode voltage-clamp OpusXpress 6000A (Molecular Devices, San José, CA) [13,14]. Continuous perfusion, mediated by Minipuls 3 peristaltic pumps (Gilson, Middleton, WI, USA), was maintained at 0.5 mL/min. Data were acquired at a membrane potential of −80 mV and sampled at 156 Hz using OpusXpress 1.10.42 (Molecular Devices) software. To increase the inward-rectifier potassium channel current at negative potentials, a high potassium concentration extracellular perfusion fluid was used (64 mM NaCl, 25 mM KCl, 0.8 mM MgCl_2_, 0.4 mM CaCl_2_, 15 mM HEPES, 1 mM ascorbic acid, adjusted to pH 7.4 with NaOH), yielding a K^+^ reversal potential of about −40 mV. Ascorbic acid prevented the spontaneous oxidation of DA. For sodium-depleted conditions, NaCl was substituted for equimolar amounts of N-methyl-D-gluconate (NMDG).

Oocytes were perfused with buffer at 1.5 mL/min for concentration-response experiments, and 1.3 mL/min for the initial (baseline and inhibition) phases of response recovery experiments. A higher, 4.5 mL perfusion rate was used during the recovery (SB269,652 washout) phase of these experiments. DA potencies at the WT, V91A, L94A, E95A, W100A, S193A, and S194A D_2_R and in the presence of NMDG were determined using repeated 25 s DA applications with intermittent washout of DA. For each oocyte, the GIRK response to a given concentration of DA was normalized to the response to the highest concentration of DA tested (1 µM for experiments with the WT D_2_R, V91A, L94A, and E95A mutants, 100 µM for W100A, 10 µM for S194A, and 300 µM for S193A).

### 2.5. Data Analysis

Electrophysiological data were analyzed in Clampfit 10.6 and MATLAB 2018b. Dose–response curves were calculated using the variable-slope sigmoidal functions in GraphPad Prism 8.

For agonist data, the following equation was used:(1)$$Y=1/\left(1+10^{\left(\log{\mathrm{EC}}_{50}-X\right)}\right)$$

For curve shift experiments, the curve shift ratio (CR) was determined as the ratio between the DA EC_50_ in the presence of a given concentration of SB269,652 and the DA EC_50_ under control conditions. To visualize the interaction between DA and SB269,652, log(CR−1) was plotted against log[SB269,652].

Inhibition data were normalized to the GIRK current response to 10 nM DA (100 nM DA in experiments with W100A and S194A mutant D_2_R; 10 µM DA with S193A mutant D_2_R) and fit to the following equation:(2)$$Y=bottom+\left(1-bottom\right)/\left(1+10^{\left(X-\log{\mathrm{IC}}_{50}\right)}\right)$$
where *Y* is the response as a fraction of 1, *bottom* is the maximal response inhibition evoked by SB269,652, and *X* is the logarithm of ligand concentration.

The kinetics of recovery of the DA response from SB269,652-mediated negative modulation were quantified by fitting monoexponential functions to individual current traces. The monoexponential function was fitted over the first 100 s following SB269,652 washout. Data points are presented as mean ± SEM throughout.

Receptor structures showing mutated residues were created using Visual Molecular Dynamics (http://www.ks.uiuc.edu/Research/vmd/; accessed on 21 December 2021) [15] and published D_2_R crystal structure data (protein data bank reference: 6LUQ) [16].

Concentration–response curves were compared by analysis of variance (F-test), testing the difference between the estimated pIC_50_ values under pre- and co-application conditions. Response recovery rates were compared using Student’s *t*-test. Values for maximal inhibition were determined as 1—*bottom* and compared against the maximal inhibition at the WT D_2_R for each receptor mutant using one-way ANOVA with Dunnett’s multiple comparisons test. The significance threshold was *p* < 0.05.

## 3. Results

We set out to characterize SB269,652 in our GIRK activation assay in *Xenopus* oocytes [14,17], heterologously expressing D_2_R and GIRK channel subunits together with the GTPase accelerating protein, RGS4.

### 3.1. Temporal Dependency of SB269,652-Mediated D_2_R Response Inhibition

We used a low DA concentration of 10 nM (~EC_30_; Figure 1B; Table 1) to evoke a control GIRK response, which was used to assess the negative D_2_R modulation imparted by the addition of SB269,652 concentrations from 3 µM to 100 µM, yielding an IC_50_ of 49.0 µM and maximal inhibition of the DA-induced current response of 66% (Figure 1C,E). This high IC_50_ contrasts with previous reports in the literature, which describe SB269,652 IC_50_s at the D_2_R (in the presence of µM concentrations of DA) in the high nM range [6,7,11]. Those previous studies used pre-incubation protocols, where SB269,652 was applied prior to DA. Our recent investigation of a distinct bivalent D_2_R ligand revealed evidence of an induced-fit binding mode, where initially rapidly reversible binding becomes virtually irreversible over the time course of a few minutes [9]. We therefore investigated whether the order of the application of SB269,652 and DA could have an impact on the inhibitory potency of the former ligand. Following 87-s pre-application of each SB269,652 concentration prior to the co-application of SB269,652 with 10 nM DA, the IC_50_ was found to be 7.4 µM, 6.6-fold lower than that observed under the co-application protocol (Figure 1D,E). Shorter or longer 50-s or 220-s pre-incubation intervals yielded similar IC_50_s (5.1 µM and 2.9 µM), which were not significantly different from the IC_50_ observed with an 87-s interval (Figure 1E). To verify the non-competitive nature of SB269,652 modulation of the D_2_R, curve shift experiments were performed. In these experiments, SB269,652 was pre-applied for 100 s prior to co-application with DA for 60 s. DA-mediated GIRK responses were normalized to the response to 1 µM DA evoked in the same cell in the absence of SB269,652. Consistent with previous reports studying other D_2_R effectors [6,7], the concentration–response curves for DA-evoked GIRK activation in the presence of increasing concentrations of SB269,652 revealed a much more limited rightward shift in DA EC_50_s than would be expected from competitive inhibition (Figure 1F,G).

### 3.2. The Effect of Pre-Application on SB269,652 Potency Is Dependent on Secondary Binding Pocket Integrity

Previous investigations have demonstrated the D_2_R secondary binding pocket to confer induced-fit binding of bitopic ligands, with mutation of SBP residues increasing ligand dissociation rates [9,18]. In addition, the SBP has been shown to play an important role in the allosteric pharmacology of SB269,652 [7,11]. We therefore tested a number of alanine mutations (Figure 2A,B; Table 1) of SBP residues in the D_2_R previously described to strongly affect SB269,652 affinity and negative cooperativity with DA [11]. At the V91A mutant, pre-application with 100 µM SB269,652 during 87 s prior to the application of 10 nM DA shifted the pIC_50_ 5.1-fold (Figure 2C; Table 2). At the D_2_R L94A mutant, pre-application of SB269,652 provided a 6.8-fold more potent inhibition compared to the co-application condition (Figure 2D; Table 2), whereas mutating the residue E95 to alanine reduced the potency shift observed upon pre-application to 2.2-fold (Figure 2E; Table 2). In experiments with the W100A mutant D_2_R, 100 nM DA was used to elicit GIRK activation, to compensate for the previously observed [9] ~10-fold higher EC_50_ of DA at this mutant receptor compared to WT (Figure 2B; Table 1). Pre-application of SB269,652 resulted in a 102.3-fold leftward shift in pIC_50_ at this mutant (Figure 2F; Table 2).

Finally, we investigated the effects of mutating the conserved serine residues, S193 and S194 in TM5, which have also been shown to play a role in the pharmacology of SB269,652 [11]. These residues form part of the OBS and are considered to contact the hydroxyl groups of DA and to stabilize TM5 through intrahelical hydrogen bonds, respectively [14,19,20,21]. As a result of the lower potencies of DA at S193A and S194A (Figure 2B; Table 1) as reported previously [12,22,23], we used 10 µM and 100 nM DA, respectively, in experiments with these mutants. In contrast to what was observed with the WT D_2_R and the other mutants, SB269,652 completely inhibited the response to 10 nM DA at the S193A mutant and displayed virtually identical potency in pre- and co-application experiments (1.1-fold potency difference; Figure 2G; Table 2), while at S194A, the maximal response inhibition by SB269,652 was sub-maximal and pre-application increased SB269,652 potency by 7.9-fold (Figure 2H; Table 2).

### 3.3. Pre-Application Slows Rate of Recovery from SB269,652 Modulation

To determine whether the increased SB269,652 potency following pre-application was related to differences in the rates of dissociation of SB269,652 from the D_2_R, we measured the kinetics of recovery of the DA-induced GIRK response from SB269,652-mediated inhibition [9,17] following either the co- or pre-application of SB269,652. Following inhibition upon co-application of SB269,652 with 10 nM DA, the removal of SB269,652 from the extracellular buffer in the continued presence of DA resulted in a swift recovery of the DA-mediated response (Figure 3A,C; blue trace and column). The pre-application of 100 µM SB269,652 for 87 s in the absence of DA, followed first by the co-application with 10 nM DA and subsequently by the removal of SB269,652 in the continued presence of DA, resulted in a slower rate of response recovery than that observed under the co-application condition (Figure 3A,C; purple trace and column). At the S193A mutant D_2_R, no effect of pre- vs. co-application was observed (Figure 3B,C).

### 3.4. Absence of Sodium Ions Reduces Pre-Application-Induced Increase in SB269,652 Potency

A previous report proposed that the NAM properties of SB269,652 were sodium-dependent [24]. To test the effect of nominally sodium-free conditions in the GIRK assay, sodium chloride in the extracellular buffer was replaced by N-methyl-D-gluconate (NMDG). While both DA and SB269,652 appeared to be more potent in the absence of sodium (Figure 4A,B; Table 1 and Table 2), pre-application of SB during 87 s induced a somewhat smaller IC_50_ shift of 3.1-fold (Figure 3B, Table 2).

## 4. Discussion

SB269,652 demonstrated an unexpectedly low potency at the WT D_2_R in the GIRK assay with an IC_50_ of 49 µM in co-application experiments. However, pre-application of SB269,652 during 50, 87, or 220 s increased the inhibitory potency of this ligand, such that the IC_50_ was 6.6- to 17-fold lower. In response recovery experiments, a full recovery of the response to 10 nM DA was observed when SB269,652 was either co- or pre-applied and subsequently washed out in the continued presence of DA. However, recovery at the WT D_2_R was slower when SB269,652 had been applied prior to DA, suggesting that the increase in potency in pre-application experiments may, at least in part, reflect a reduction of the dissociation rate of the ligand under these conditions.

The greatest effects of D_2_R mutations to reduce this increase in SB269,652 potency upon pre-application was observed with E95A and S193A. Interactions with E95 have consistently been found to be important for the allosteric actions of SB269,652 and for the binding of other allosteric D_2_R ligands [25,26]. Notably, in the present experiments, an E95A mutation reduced the shift in potency upon SB269,652 pre-application to about two-fold. A S193A mutation appeared to abolish the effect of pre-application on SB269,652 potency and furthermore allowed for a complete inhibition of the DA-induced response at high concentrations of the ligand, while at the WT D_2_R, as well as at the other mutant receptors, inhibition levelled off at submaximal levels, in agreement with the allosteric properties and limited negative cooperativity of SB269,652 [6,7]. S193A mutation has been reported to increase the negative cooperativity of SB269,652, presumably by a better accommodation of the cyano group on the primary pharmacophore and consequently improved interactions with E95A, as suggested by previous molecular dynamics simulations [11]. The increase in the negative cooperativity may be in agreement with our observation of a complete, rather than incomplete, inhibition of the DA-induced GIRK response by SB269,652 at this mutant. Conversely, the V91A and E95A mutations have been suggested to perturb ligand interactions with the SBP, consequently decreasing negative cooperativity [7,8,11]. This would be in line with the reduction in maximal SB269,652-mediated inhibition in our experiments with these two mutants (Figure 2C,E; Table 3).

The higher potency of and slower recovery rate from SB269,652 inhibition at the WT D_2_R when oocytes were pre-incubated with the compound prior to DA application suggests that SB269,652 binds in a different manner under these circumstances, compared to when the ligand is co-applied with DA. In agreement with the similar potency between pre- and co-applied SB269,652 at the D_2_R S193A mutant, the recovery rates were also similar under both application protocols when tested with this mutant. These findings are reminiscent of those of another recent investigation of ours, where we found evidence of an induced-fit-like binding mechanism [27] of the bivalent, competitive D_2_R ligand SV-III-130, which exhibited a time-dependent transition into an irreversibly-bound state [9]. Another mechanism which could produce this type of behavior is “conformational selection”: preferential binding to transient, higher-affinity receptor states, leading to the enrichment of those conformations. Indeed, conformational selection has been reported for orthosteric ligands at the D_3_R [28] and for allosteric muscarinic receptor ligands [29].

In agreement with the present findings, it has been proposed that SB269,652 may interact differentially with the D_2_R depending on whether the OBS is occupied by another ligand [5]. An intriguing possibility is that the higher-affinity binding mode favored by pre-application involves the binding of the THIQ moiety of SB269,652 to the OBS, and is dependent on an interaction with S193. The finding that E95A decreased the impact of pre-application on SB269,652 potency suggests that this binding mode would also be dependent on an intact SBP, which presumably both confers allosteric modulation and positioning of the THIQ core for an optimal interaction with the OBS [10,30]. Conversely, the low-potency inhibition by SB269,652 which is observed upon simultaneous application with DA may result mainly from allosteric interactions of the indole-2-carboxamide fragment with the SBP, with little or no involvement of the OBS.

We included W100 in our alanine mutagenesis experiments because of the large impact of this residue on the accessibility of the OBS from the SBP [31]. Furthermore, X-ray crystallography and computational studies suggest a close interaction of W100 with I184 in ECL2 and with L94 [9,18], both of which have been proposed to interact with SB269,652 [11,24]. We previously found a W100A mutation to abolish induced-fit binding of SV-III-130 to the D_2_R [9]. However, somewhat surprisingly, the W100A mutation drastically increased the pre-application effect on SB269,652 potency, from ~6.6-fold to ~102.3-fold. This increase appears to result both from a reduced potency in co-application experiments and from increased potency in pre-application experiments. It thus appears that W100 contributes positively to binding affinity at the SBP in the binding mode favored by co-application experiments, perhaps by sandwiching the indole-2-carboxamide fragment between itself and I184 [9]. Conversely, the replacement of the bulky indole sidechain with a methyl in W100A might better accommodate the ligand in the presumed second binding mode favored by pre-application.

Draper-Joyce et al. reported a loss of D_2_R NAM properties of SB269,652 in the absence of extracellular sodium [24]. Typically, GPCRs harbor a sodium ion between TM1–3 and TM6–7, with a conserved aspartate residue in TM2 being important for sodium ion coordination, agonist potency, and constitutive receptor activity [32,33]. Thus, assuming that sodium depletion by NMDG replacement is likely to affect the conformations of the OBS and the SBP, the properties of SB269,652 might be expected to change in our functional experiments. The present results suggest that the pre-application effect on SB269,652 potency is reduced when sodium is replaced by NMDG (a 3.1-fold vs. a 6.6-fold potency increase upon pre-application; compare Figure 4B with Figure 1E), although this reduction is smaller compared to that caused by E95A mutation (2.2-fold potency increase upon pre-application; see Figure 2E).

Previous cell-based assays of SB269,652 activity typically involved pre-incubation periods of about 30 min. Although little difference was observed between pre-incubation periods of 50 s, 87 s, and 220 s in our experiments, we cannot exclude the possibility that a longer pre-incubation with SB269,652 would have revealed a higher potency of the ligand. An association rate analysis of SB269,652 at the D_2_R was complicated by the low potency in our assay, which prevented us from estimating association rate constants of this compound as we previously did for a number of D_2_R antagonists [17]. Further experimentation will be required to elucidate the molecular details of the two binding modes of SB269,652 at the D_2_R suggested by the present data. In particular, it would be interesting to study the effects of SB269,652 in time-resolved readouts of ligand binding and of other signaling pathways downstream of D_2_R, such as β-arrestin recruitment and cAMP accumulation. On the basis of our present results, we would expect to observe a corresponding increase in SB269,652 affinity and functional potency also in such assays when the ligand is pre-incubated with the D_2_R instead of being co-applied with DA.

## 5. Conclusions

The inhibitory potency of SB269,652 was higher when pre-incubated with D_2_R-expressing oocytes before DA addition compared to when the compound was co-applied with DA. In addition, the recovery of the D_2_R-mediated response to DA upon the washout of SB269,652 was slower in pre-incubation experiments. This suggests that SB269,652 induces or selects a D_2_R conformation with higher affinity for this ligand upon binding. Since this mechanism is dependent on the order of application of DA and SB269,652, it is presumably competitive in nature or disfavored by simultaneous DA binding. Mutagenesis experiments suggest that both the SBP and the OBS of D_2_R participate in this mechanism, since the effect of pre-incubation to increase SB269,652 potency was strongly reduced by both E95A and S193A mutation. This information is another piece of the puzzle of understanding the complex nature of allosteric D_2_R inhibition provided by a bitopic, partly orthosteric ligand.

## Figures and Tables

**Figure 1 biomedicines-10-00022-f001:**
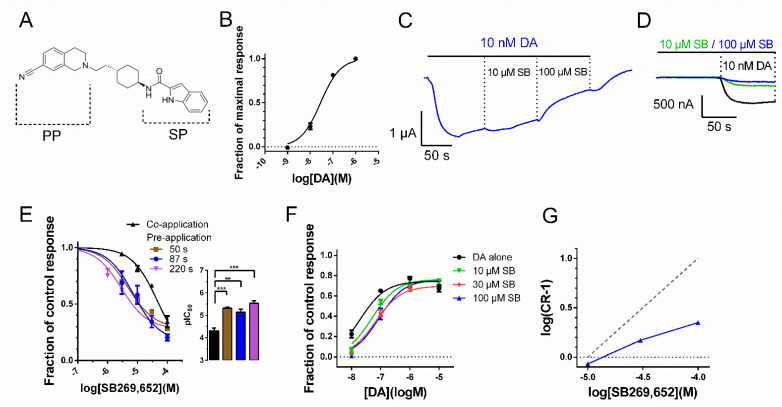
Pre-application increases SB269,652 potency at D_2_R. (**A**) Structure of SB269,652, indicating the THIQ primary pharmacophore (PP) and the indole-2-carboxamide secondary pharmacophore (SP). (**B**) Concentration-response relationship for DA at the WT D_2_R. Equation (1) was fit to the data as described in the Methods. (**C**) Representative trace showing the GIRK current response to application of 10 nM DA and subsequent addition of 10 and 100 µM SB269,652. (**D**) Representative traces from the same oocyte showing GIRK current responses to application of 10 nM DA following 87-s pre-application of 10 µM (green trace) or 100 µM (blue trace) SB269,652, or control buffer solution (black trace). (**E**) Concentration-response relationships for SB269,652, co-applied with 10 nM DA (black; pIC_50_ = 4.31 ± 0.14 [49.0 µM], *n* = 3–6) or pre-applied alone during 50 s (brown curve, pIC_50_ = 5.31 ± 0.06 [4.9 µM], *n* = 6), 87 s (blue curve, pIC_50_ = 5.13 ± 0.16 [7.4 µM], *n* = 4–7), or during 220 s (purple curve, pIC_50_ = 5.54 ± 0.11 [2.9 µM], *n* = 4). Data were normalized within cells to the response to 10 nM DA applied without SB269,652. Equation (2) was fit to the inhibition data as described in the Methods. Asterisks indicate statistical significance between the fitted pIC_50_ parameters for the co- and pre-incubation conditions: **, *p* < 0.01; ***, *p* < 0.001, F-test. There were no significant differences when pIC_50_ values estimated under the three different pre-application conditions were tested individually against each other. (**F**) Curve shift data for DA in the absence or presence of increasing concentrations of SB269,652. Curves show fits to Equation (1). DA pEC_50_s were estimated as 7.70 ± 0.08 under control conditions (black curve; *n* = 7–8) and as 7.30 ± 0.08 (green curve; *n* = 5–6), 7.17 ± 0.07 (red curve; *n* = 6), and 7.06 ± 0.11 (blue curve; *n* = 4) in the presence of 10, 30, or 100 µM SB269,652, respectively. Data are displayed as mean ± SEM. (**G**) Schild plot showing the logarithm of the curve shift ratio (CR)—1 as a function of SB269,652 concentration. The dotted diagonal line represents the unity slope expected of a competitive antagonist. Data represent means of EC_50_ estimates from the curve fits in (**F**).

**Figure 2 biomedicines-10-00022-f002:**
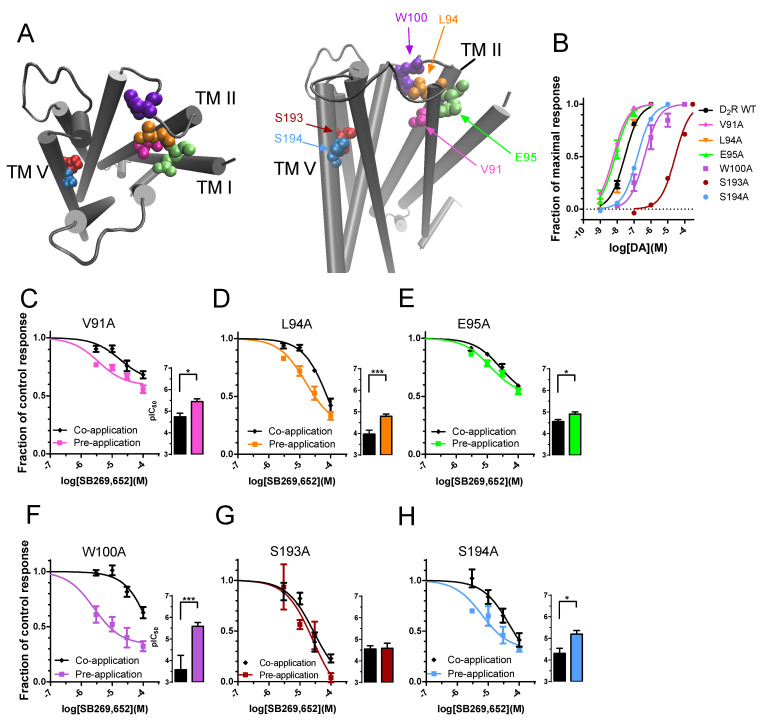
Mutation of critical residues in the D_2_R OBS and SBP reduce the effect of pre-application on SB269,652 potency. (**A**) Schematic representation of D_2_R structure showing the location of the mutated residues in the OBS and SBP. (**B**) DA potencies at the WT D_2_R and at the SBP and OBS mutants. Equation (1) was fit to the agonist data as described in the Methods section. See Table 1 for EC_50_ values. (**C**) Concentration–response relationships for SB269,652 at the D_2_R V91A mutant, co-applied with 10 nM DA (black, *n* = 7–9) and pre-applied during 87 s (pink, *n* = 5). (**D**) Concentration–response relationships for SB269,652 at the D_2_R L94A mutant, co-applied with 10 nM DA (black, *n* = 5–7) and pre-applied during 87 s (orange, *n* = 5). (**E**) Concentration–response relationships for SB269,652 at the D_2_R E95A mutant, co-applied with 10 nM DA (black, *n* = 9–10) and pre-applied during 87 s (green, *n* = 6). (**F**) Concentration–response relationships for SB269,652 at the D_2_R W100A mutant, co-applied with 100 nM DA (black, *n* = 5–7) and pre-applied during 87 s (purple, *n* = 6). (**G**) Concentration–response relationships for SB269,652 at the D_2_R S193A mutant, co-applied with 10 µM DA (black, *n* = 6) and pre-applied during 87 s (dark red, *n* = 3). (**H**) Concentration–response relationships for SB269,652 at the D_2_R S194A mutant, co-applied with 100 nM DA (black, *n* = 4–5) and pre-applied during 87 s (blue, *n* = 3). Equation (2) was fit to the inhibition data as described in the Methods section. See Table 2 for pIC_50_ values. Data are displayed as mean ± SEM. Asterisks indicate statistical significance: *, *p* < 0.05; ***, *p* < 0.001, F-test.

**Figure 3 biomedicines-10-00022-f003:**
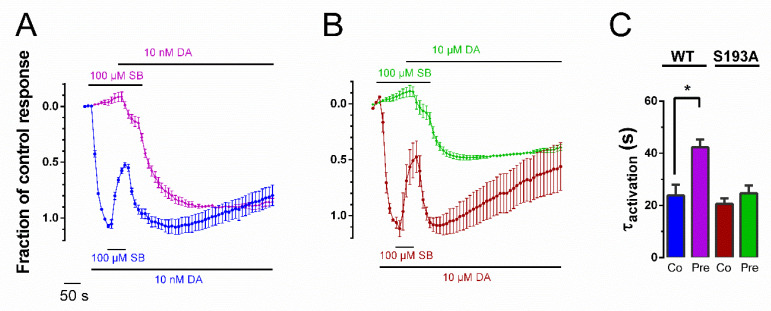
Pre-application slows rate of response recovery from SB269,652 modulation. (**A**) Response recovery from SB269,652 modulation at the WT D_2_R. Blue trace: 50-s application of 10 nM DA followed by 43-s co-application of 100 µM SB269,652 with 10 nM DA and subsequent washout of SB269,652 in the continuous presence of 10 nM DA (*n* = 5). Pink trace: 87 s pre-application of 100 µM SB269,652 followed by co-application of 100 µM SB269,652 with 10 nM DA and subsequent washout of SB269,652 in the continuous presence of 10 nM DA (*n* = 3). The time course of recovery of the DA response was quantified by fitting an exponential function to the response recovery time course of individual traces during the initial 100 s after washout of SB269,652, yielding the time constant τ. Data is shown normalized to the maximal response evoked by 10 nM DA in the same cell. (**B**) Response recovery from SB269,652 modulation at the D_2_R S193A mutant under conditions corresponding to those in (**A**), with the red trace representing co-application (*n* = 3) and the green trace representing pre-application (*n* = 3) of SB269,652 with DA. (**C**) Mean values of τ for response recovery time courses for the pre- and co-application conditions for the WT and S193A mutant D_2_R. Asterisk indicates statistical significance for pre- and co-application conditions for the WT D_2_R: *, *p* ≈ 0.018, Student’s unpaired *t*-test. For the D_2_R S193A mutant, τ was not significantly different between pre- and co-application conditions (*p* ≈ 0.330). Data are displayed as mean ± SEM.

**Figure 4 biomedicines-10-00022-f004:**
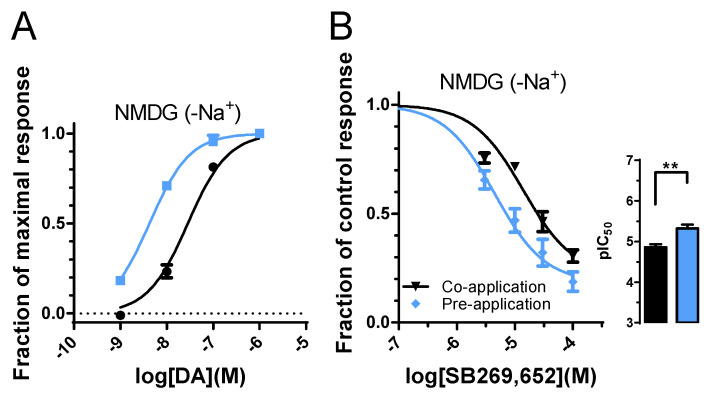
Concentration–response relationship of SB269,652 at the D_2_R under nominally sodium-free conditions. (**A**) Concentration–response relationships for DA at the WT D_2_R under control conditions (black, same as shown in Figure 1B, *n* = 8), and under sodium-free conditions (blue, *n* = 3). See Table 1 for pEC_50_ values. (**B**) Concentration–response relationships for SB269,652 at the D_2_R, co-applied with 10 nM DA (black, *n* = 5–8) and pre-applied during 87 s (blue, *n* = 5). See Table 2 for pIC_50_ values. Data are displayed as mean ± SEM. Asterisks indicate statistical significance: **, *p* < 0.01, F-test.

**Table 1 biomedicines-10-00022-t001:** DA potencies at the WT D_2_R (including the sodium-free condition; NMDG) and mutants.

D_2_R Construct	pEC_50_ ± SEM	*n*
WT	7.55 ± 0.04	8
V91A	8.23 ± 0.02	4
L94A	7.55 ± 0.07	4
E95A	8.12 ± 0.08	3
W100A	6.38 ± 0.11	5
S193A	4.56 ± 0.05	3
S194A	6.80 ± 0.02	3
WT, NMDG	8.37 ± 0.03	3

**Table 2 biomedicines-10-00022-t002:** SB269,652 potencies at the WT and mutant D_2_R, including the sodium-free condition (NMDG).

D_2_R Construct/ Condition	Co-Application pIC_50_ ± SEM	Co-Application IC_50_ (µM)	*n*	Pre-Application pIC_50_ ± SEM	Pre-Application IC_50_ (µM)	*n*	Fold Change Pre/Co-Application	Co- vs. Pre-Application (*p*-Value)
WT	4.31 ± 0.14	49.0	3–6	5.13 ± 0.16	7.4	4–7	6.6	0.0023
V91A	4.74 ± 0.21	18.2	7–9	5.45 ± 0.15	3.5	5	5.1	0.023
L94A	3.97 ± 0.23	107.2	5–7	4.80 ± 0.11	15.8	5	6.8	0.0004
E95A	4.56 ± 0.10	27.5	9–10	4.91 ± 0.10	12.3	6	2.2	0.0349
W100A	3.58 ± 0.82	263.0	5–7	5.59 ± 0.21	2.6	6	102.3	0.0001
S193A	4.52 ± 0.19	30.2	6	4.58 ± 0.28	26.3	3	1.1	0.9236
S194A	4.30 ± 0.29	50.1	4–5	5.20 ± 0.19	6.3	3	7.9	0.0156
WT/NMDG	4.86 ± 0.08	13.8	5–8	5.35 ± 0.12	4.5	5	3.1	0.0026

**Table 3 biomedicines-10-00022-t003:** Comparison between the effects of mutations of D_2_R OBS and SBP on SB269,652 negative cooperativity in a previous report [11] and maximal inhibition under pre-application conditions in the present study.

D_2_R Construct	Negative Cooperativity of SB269,652 Interaction with DA [11] ^a^	Maximal Response Inhibition (Present Study)
WT	1.23 ± 0.14 (0.54 ± 0.11) ^b^	0.83 ± 0.07
V91A	↓ 0.48 ± 0.16 *	↓ (0.41 ± 0.03) ***
L94A	↓ 0.70 ± 0.08 *	↔ (0.76 ± 0.06)
E95A	↓ 0.32 ± 0.14 *	↓ (0.48 ± 0.03) **
W100A	*n*/a	↔ (0.66 ± 0.05)
S193A	↑ (1.47 ± 0.15 *) ^b^	↑ (1.18 ± 0.24) *
S194A	↓ 0.52 ± 0.09 *	↔ (0.68 ± 0.06)

^a^ log values from pERK assay (data in parentheses from radioligand binding—functional S193A data was not provided). Data reprinted from reference [11] with permission from Elsevier. ^b^ From binding experiments. Functional data was not provided. Asterisks indicate statistical significance compared to the WT D2R: *, *p* < 0.05; **, *p* < 0.01; ***, *p* < 0.001; one-way ANOVA with Dunnett’s multiple comparisons test. Arrows indicate the direction of change of negative cooperativity and maximal response inhibition, respectively; ↑, increase; ↓, decrease; ↔, no significant change. *n*/a, data not available.

## Data Availability

The data that support the findings of this study are available from the corresponding author upon reasonable request.

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
