# Peer review of "Evidence for Two Modes of Binding of the Negative Allosteric Modulator SB269,652 to the Dopamine D2 Receptor"

_biomedicines, 2021, doi:10.3390/biomedicines10010022_

Round 1
Reviewer 1 Report
Agren and Sahlholm report about new studies with SB269,652, the first negative allosteric modulator for the dopamine D2 receptor. In this manuscript they describe interesting results with an electrophysiological method using oocytes from African clawed toad. They show different setups (co- and pre-application of dopamine and SB) and demonstrate their influence on potency using wild-type D2R and various D2R mutants. Mutations were chosen to investigate crucial amino acid regions for orthosteric binding site (OBS) and secondary binding pocket (SBP). I recommend the manuscript to be considered for acceptance in Biomedicines after addressing the following points (major revision):
- I don’t see any reason for commas in the title. I would change it to “Evidence for two modes of binding of the negative allosteric modulator SB269,652 to the dopamine D2 receptor
- I don’t like the term, e.g. “in D2R” or “to D2R”. I would use “in/to D2Rs” or “in/to the D2R” but prefer the latter.
- The resolution of the figures should be better. Especially the graphs in Figure 1 & 2
- Could you please specify the y-axis in your graphs a little more? “Normalized response” is very common.
- Page 7 line 244f: I think at this point the authors should compare these data with the ones from Figure 1E a little more in detail or at least refer to it when it will be discussed later in the discussion chapter.
- I think it would be helpful to show an additional table where the results from this study were compared with the already published data for SB269,652.
- The English is fine for the most part. However, I would suggest that the manuscript be proofread by a native speaker. Commas are partly wrong.
- Page 7 line 277: I don’t think you need the hyphen in “of-“
- Line 282f: “another” and “investigations” doesn’t work together here.
- I can’t find an explanation for “PP” and “SP” (Figure 1A & text (e.g. line 273).
- It would be interesting to see the SB269,652 effect with longer pre-incubation periods as literature has incubation periods of 30 minutes. This could be done only for selected setups.
- Line 334-337: This sentence is a little too long and written in a pretty complicated way. Please rephrase the sentence.
Author Response
Agren and Sahlholm report about new studies with SB269,652, the first negative allosteric modulator for the dopamine D2 receptor. In this manuscript they describe interesting results with an electrophysiological method using oocytes from African clawed toad. They show different setups (co- and pre-application of dopamine and SB) and demonstrate their influence on potency using wild-type D2R and various D2R mutants. Mutations were chosen to investigate crucial amino acid regions for orthosteric binding site (OBS) and secondary binding pocket (SBP). I recommend the manuscript to be considered for acceptance in Biomedicines after addressing the following points (major revision):
Our response: We thank the reviewer for the thorough reading of our manuscript and for the helpful comments, which we have tried to address below.
- I don’t see any reason for commas in the title. I would change it to “Evidence for two modes of binding of the negative allosteric modulator SB269,652 to the dopamine D2 receptor
Our response: We have now removed the commas from the title.
- I don’t like the term, e.g. “in D2R” or “to D2R”. I would use “in/to D2Rs” or “in/to the D2R” but prefer the latter.
Our response: We now use “in/to/at the D2R” throughout, and a plural “s” with “D2/3Rs”.
- The resolution of the figures should be better. Especially the graphs in Figure 1 & 2
Our response: All figures are now provided in higher resolution.
- Could you please specify the y-axis in your graphs a little more? “Normalized response” is very common.
Our response: We now use “Fraction of maximal response” for agonist concentration-response curves and “Fraction of control response” for graphs illustrating negative modulation by SB269,652.
- Page 7 line 244f: I think at this point the authors should compare these data with the ones from Figure 1E a little more in detail or at least refer to it when it will be discussed later in the discussion chapter.
Our response: We now explicitly refer to Figure 1E, Figure 2E, and Figure 4B, as well as the fold changes in SB269,652 potency between co- and pre-application conditions under the respective conditions, in the Discussion (page 11 lines 378-381). We felt that this comparison belongs in the Discussion, rather than in the Results section.
- I think it would be helpful to show an additional table where the results from this study were compared with the already published data for SB269,652.
We now provide a table comparing previously published functional data on negative cooperativity for SB269,652 at several mutants with the maximal inhibition by SB269,652 at the same mutants in the present study (page 9, table 3).
- The English is fine for the most part. However, I would suggest that the manuscript be proofread by a native speaker. Commas are partly wrong.
Our response: We have now gone over the whole text and removed/corrected a number of commas.
- Page 7 line 277: I don’t think you need the hyphen in “of-“
Our response: We have removed this hyphen.
- Line 282f: “another” and “investigations” doesn’t work together here.
Our response: We have removed the plural “s” in “investigations”.
- I can’t find an explanation for “PP” and “SP” (Figure 1A & text (e.g. line 273).
Our response: We now define “PP” and “SP” as “primary pharmacophore” and “secondary pharmacophore”, respectively, in the legend of Figure 1. In the main text, we now use the unabbreviated forms throughout for clarity.
- It would be interesting to see the SB269,652 effect with longer pre-incubation periods as literature has incubation periods of 30 minutes. This could be done only for selected setups.
Our response: The two-electrode technique used in our electrophysiology assay is somewhat invasive in that the two electrodes penetrate the oocyte membrane, which may lead to deterioration of the cell preparation over time. Thus, we used a more limited pre-incubation period in order to obtain stable, high-quality recordings. However, we now provide additional data obtained with a 220-s pre-incubation interval for the WT D2R. SB269,652 potency in these experiments was similar to that observed with the shorter pre-incubation periods of 50 and 87 s (Page 5, Figure 1E).
- Line 334-337: This sentence is a little too long and written in a pretty complicated way. Please rephrase the sentence.
Our response: We have now rephrased and broken up this sentence into three.
Reviewer 2 Report
The study by Ågren and Sahlholm employs structure/function studies and electrophysiologic recordings in Xenopus oocytes to study the interaction of a negative allosteric modulator of dopamine D2 receptor with the orthostheric and allosteric sites. This is a timely study as this type of compound could have beneficial clinical effects as it could have less unwarranted side effects compared to more classical antipsychotics. Furthermore, the mechanism of action of this type of ligand remains somewhat controversial warranting studies like this. The paper is well written, and the study is well designed with carefully employed assays. Several different ways of exploring the topic are employed to support the general conclusion. A strong advantage of the study it the use of an assay with excellent time resolution to provide interesting new mechanistic information.
Some points that will strengthen the study:
It appears that most previous studies using other assays have only used pre-incubation. It is probably beyond the scope of this study to employ these assays including binding, cAMP, arrestin-based assays and see if they also find a similar difference in affinity when comparing with co-incubation. However, more discussions on what would be expected if they were performed could be included.
It also appears other allosteric compounds including those used in ref 8 exist that does not have a bimodal interaction with the allosteric and the orthosteric site. If any compounds that only interact with the allosteric site are available these should be included in these studies to help dissect the role of the two sites to gain a better understanding of mechanistic aspects of this process.
For some mutants it appears that only partial vs full inhibition is observed. The authors should include some discussion on this. This also brings up, that competition for the same sites is suggested and discussed. That should be tested by doing dose response curves in the presence of the other compound to test if only a rightward shift is observed or if the interaction is non-competitive and also the maximum response is affected. Experiments like that on the WT together with what they find on the mutants will help to clarify and support their hypothesis.
Also, along these lines the discussion on the various mutants and the differences in the magnitude of effect should be expanded. Do those findings tell something about the molecular determinants of the binding site?
It appears no statistical analysis have been used to determine if differences are significant or not. This should be included.
Author Response
The study by Ågren and Sahlholm employs structure/function studies and electrophysiologic recordings in Xenopus oocytes to study the interaction of a negative allosteric modulator of dopamine D2 receptor with the orthostheric and allosteric sites. This is a timely study as this type of compound could have beneficial clinical effects as it could have less unwarranted side effects compared to more classical antipsychotics. Furthermore, the mechanism of action of this type of ligand remains somewhat controversial warranting studies like this. The paper is well written, and the study is well designed with carefully employed assays. Several different ways of exploring the topic are employed to support the general conclusion. A strong advantage of the study it the use of an assay with excellent time resolution to provide interesting new mechanistic information.
Our response: We thank the reviewer for the positive comments and for the interesting suggestions, most of which we have tried to implement below.
Some points that will strengthen the study:
It appears that most previous studies using other assays have only used pre-incubation. It is probably beyond the scope of this study to employ these assays including binding, cAMP, arrestin-based assays and see if they also find a similar difference in affinity when comparing with co-incubation. However, more discussions on what would be expected if they were performed could be included.
Our response: We have now added some discussion about what would be expected from time-resolved studies of ligand binding, cAMP accumulation, and arrestin recruitment (page 11, lines 390-396).
It also appears other allosteric compounds including those used in ref 8 exist that does not have a bimodal interaction with the allosteric and the orthosteric site. If any compounds that only interact with the allosteric site are available these should be included in these studies to help dissect the role of the two sites to gain a better understanding of mechanistic aspects of this process.
Our response: This is a very interesting idea. However, the compounds (fragments of SB269,652) which only interact with the allosteric/secondary binding site are reported to have very low affinity with KBs of 6 and 46 µM, while the KB of SB269,652 itself was given as 776 nM (see Fig. 7 in ref. 8). Given that the potency of SB269,652 is already very low in our assay, we would expect difficulties in obtaining useful data with the allosteric fragments.
For some mutants it appears that only partial vs full inhibition is observed. The authors should include some discussion on this. This also brings up, that competition for the same sites is suggested and discussed. That should be tested by doing dose response curves in the presence of the other compound to test if only a rightward shift is observed or if the interaction is non-competitive and also the maximum response is affected. Experiments like that on the WT together with what they find on the mutants will help to clarify and support their hypothesis.
Our response: The decreased and increased maximal inhibition by SB269,652 observed at the V91A and E95A mutants on the one hand and the S193A mutant on the other corresponds well with the reductions and increases in negative cooperativity of the compound at these mutants, as previously reported in the literature. We have added a new table and some discussion about this (page 9, Table 3; page 10, lines 327-330).
We have also included curve shift data for the WT D2R showing that, in agreement with previous studies, SB induces a rightward shift in the DA concentration-response curve which is considerably more limited than what would be expected from a competitive antagonist (thus being more in agreement with an allosteric effect), with no apparent effect on the maximum DA-induced response (page 4, lines 168-175; page 5, Figure 1F, G).
Also, along these lines the discussion on the various mutants and the differences in the magnitude of effect should be expanded. Do those findings tell something about the molecular determinants of the binding site?
Our response: We have now expanded the discussion about the mutants, as is also mentioned in our response to the point above. We also provide a table (page 9, Table 3) comparing the maximal response inhibition to negative cooperativity data from the previous study by Draper-Joyce et al. (Biochem Pharmacol. 2018).
It appears no statistical analysis have been used to determine if differences are significant or not. This should be included.
Our response: We have now included statistical analyses of the shifts in SB269,652 potency upon pre-incubation vs. co-incubation with DA (Figure 1, Figure 2, Figure 4), as well as the differences in maximal inhibition between the mutants (page 9, Table 3).
Round 2
Reviewer 1 Report
The authors have addressed all points accordingly. The graphs in Figure 1 & 2 are fine now but the structure in Figure 1A is even worse in the revised version. Please provide a sharp version of that chemical structure.
The revised version of the manuscript can be published as is after this minor revision.
Author Response
The authors have addressed all points accordingly. The graphs in Figure 1 & 2 are fine now but the structure in Figure 1A is even worse in the revised version. Please provide a sharp version of that chemical structure.
Our response: We are sorry for the low resolution of this panel. We believe that this may have been due to export settings when converting the image to .pdf format, since the source image was reasonably high resolution and looked fine (at least to us) in the Word document. We now provide a new figure which looks sharp both in Word and (after optimizing export/print settings in Word) in the .pdf.
Reviewer 2 Report
The authors have fully addressed my suggestions.
Author Response
Our response: We are happy that we could address the reviewer's helpful suggestions.